# Long-Term Sorption and Solubility of Zirconia-Impregnated PMMA Nanocomposite in Water and Artificial Saliva

**DOI:** 10.3390/ma13173732

**Published:** 2020-08-24

**Authors:** Saleh Zidan, Nikolaos Silikas, Julfikar Haider, Julian Yates

**Affiliations:** 1Dentistry, School of Medical Sciences, University of Manchester, Manchester M13 9PL, UK; nikolaos.silikas@manchester.ac.uk (N.S.); julian.yates@manchester.ac.uk (J.Y.); 2Department of Dental Materials, Faculty of Dentistry, Sebha University, Sebha, Libya; 3Department of Engineering, Manchester Metropolitan University, Manchester M1 5GD, UK; j.haider@mmu.ac.uk

**Keywords:** denture base, high-impact PMMA, zirconia (ZrO_2_), nanocomposite, sorption, solubility, artificial saliva

## Abstract

Exposure of denture base acrylic resins to the oral environment and storage media for extended periods of time results in sorption of saliva or water, leading to a reduction in physical properties and thus clinical service life. The purpose of this in vitro study was to assess the sorption and solubility of high-impact heat-polymerised denture base acrylic resin (HI PMMA) impregnated with zirconia nanoparticles after being stored for 180 days in distilled water (DW) and artificial saliva (AS). The specimens were divided into six groups for each storage medium, according to the concentration of zirconia nanoparticles (0, 1.5, 3.0, 5.0, 7.0, and 10.0 wt.%). Data were statistically analysed for sorption and solubility using one-way and two-way ANOVA statistical tests. Sorption in DW and AS for all groups containing zirconia showed sorption values lower than the control group at 90 days, though not significantly different (*p* > 0.05) compared to the control group. For both the DW and AS groups, the lowest solubility value was measured in the group containing 3 wt.% zirconia, however, there was no significant difference compared to the control group except when observing 10 wt.% zirconia in AS, which showed a significantly higher solubility (*p* < 0.05). High-impact PMMA, impregnated with low concentrations of ZrO_2_, showed the lowest sorption and solubility in both media, but was not significantly different compared to pure HI PMMA.

## 1. Introduction

Since the beginning of the twentieth century, poly-methyl methacrylate (PMMA) acrylic resin has been the material of choice for constructing denture bases [1]. PMMA resin has been reinforced with butadiene-styrene to develop high-impact (HI) PMMA in an attempt to improve its physical and mechanical properties. These materials are provided in a powder–liquid form and are processed in the same way as other conventional heat-polymerised methyl methacrylate resins [2]. Within the oral environment, the physical properties of PMMA undergo rapid changes with time and this influences the mechanical properties [3]. Water sorption and solubility are major problems that can affect the durability of the denture bases [4]. In relation to general aesthetics, in the long term, staining affects the gloss surface and the shine of the denture, which are of concern to both patient and dentist [3].

Poly-methyl methacrylate (PMMA) absorbs water slowly over a period time in the oral environment [3,5,6], because of the polarity of the resin molecules. By forcing the polymer chains apart, the absorbed water spreads between the macromolecules of the material by diffusion [4], causing an expansion that may compensate for the polymerisation shrinkage that occurs during fabrication of the heat-cured denture bases [7]. The amount of water sorption into a polymeric structure is influenced by the degree of conversion, the cross-link density and hydrophilicity of the polymeric network [8]. Additionally, water absorbed into the PMMA material can act as a plasticiser, causing it to become soft, which can result in a physical change of the polymer networks. This change can lead to a decrease in mechanical properties, such as transverse strength, hardness and fatigue limit, and thus possibly increases the potential for fractures [6,7,9,10]. High water sorption rates tend to affect the material properties and reduce the clinical lifespan of a denture within the oral cavity. Therefore, it is important to use materials with the lowest possible water sorption rates [11]. According to British Standard (BS EN ISO 20795-1), water sorption should not exceed 32 µg/mm^3^ for heat-cured acrylic resin materials [12].

Solubility represents the fraction of material dissolved from a polymer [13]. In general, denture base acrylic resins have low solubility, and the little that occurs is a result of the leaching out of traces of unreacted monomer and water-soluble additives, plasticisers and initiators into the oral fluids [7]. However, these monomers sometimes produce a soft tissue reaction [4]. According to the British Standard (EN ISO) specification No. 20795, the solubility of denture bases should not exceed 1.6 µg/mm^3^ for heat-cured acrylic resin materials [12].

Pfeiffer and Rosenbauer conducted a study to evaluate water sorption and solubility of denture base resins (Sinomer: heat-polymerised, modified methacrylate; Polyan: thermoplastic, modified methacrylate; Promysan: thermoplastic, en-terephthalate; Microbase: microwave polymerised, polyurethane-based, and Paladon 65: heat-polymerised, methacrylate, control group). The findings showed that thermoplastic polymerisation (Polyan) resulted in lower water sorption than conventional heat-cured acrylic resin and other groups. However, water solubility was not significantly different among the groups [14]. Zuo et al. evaluated organic–inorganic hybrid coated denture base surfaces for water sorption and solubility. They reported that water uptake and solubility were reduced, due to the application of the coating [9]. Asar et al. also demonstrated that reinforcing conventional heat-polymerised acrylic resin with different metal oxide fillers (Al_2_O_3_, TiO_2_ and ZrO_2_) significantly decreased water sorption and solubility, particularly with the use of ZrO_2_ [7].

Recent literature shows that the addition of nanoscale reinforcing agents with polymer materials produces new mechanical and physical properties, creating a new class of nanocomposite. In dentistry, many attempts have been made to create and improve PMMA with the addition of different nanosized fillers [15], and their physical and mechanical properties evaluated. However, there is no study in the literature investigating the physical properties (e.g., water sorption and solubility) of HI heat-cured denture base resins reinforced with zirconia nanoparticles.

In our previous study, it was found that a small amount of zirconia addition to HI-PMMA can further improve the mechanical strength [16]. However, if the sorption and solubility of this new material is more than the standard recommended values, the mechanical properties would be affected. Therefore, it is important to understand the sorption and solubility characteristics of the HI PMMA-Zirconia nanocomposite from material and clinical performance point of views. The aim of this study was to evaluate the effect of adding zirconia nanoparticles at different concentrations to high-impact PMMA on sorption and solubility when stored in distilled water (DW) and artificial saliva (AS) for up to 180 days.

The hypotheses to be tested were that there would be no significant change in (i) sorption and (ii) solubility for HI PMMA nanocomposite experimental groups compared to the pure HI PMMA (control group) resin.

## 2. Materials and Experimental Method

### 2.1. Materials

A commercially available Metrocryl HI denture base powder, (PMMA, poly-methyl methacrylate) and Metrocryl HI (X-Linked) denture base liquid (MMA, methyl methacrylate) (Metrodent Limited, Huddersfield, UK) were selected as the denture base material. Yttria-stabilised zirconia (ZrO_2_) nanoparticles (Sky Spring Nano materials, Inc., Houston, TX, USA) were chosen as the inorganic filler agent for fabricating nanocomposite denture specimens.

### 2.2. Preparation of Specimens

Zirconia nanoparticle surfaces were treated with 7 wt.% silane coupling agent (3-trimethoxysilyl propyl methacrylate; product no. 440159, Sigma-Aldrich, Gillingham, UK) as detailed in [16]. According to a previous study [16], the most appropriate weight percentages of zirconia nanoparticles were used in this study: 0 wt.% (control), 1.5, 3.0, 5.0, 7.0, and 10.0 wt.%.

The silane-treated zirconia nanoparticles were added to the acrylic resin monomer and mixed by hand using a stainless-steel spatula to ensure all the powder was uniformly distributed within the resin monomer and to avoid any aggregation. The HI acrylic resin powder was then added to the liquid, and mixing continued until a consistent mixture was obtained, according to the manufacturer’s instructions. When the mixture reached a dough-like stage that was suitable for handling, it was packed into a mould by hand. The mould was then closed and placed in a hydraulic press (Sirio P400/13045) under a pressure of 15 MPa and then excess mixture was removed from the periphery of the mould. The mould was immersed in a curing bath for 6 h of polymerisation at temperature 95 °C, and then the mould was removed from the curing bath and cooled slowly for 30 min at room temperature. The mould was then opened, and the specimens were removed. The specimens were then trimmed using a tungsten carbide bur, ground with an emery paper and polished with pumice powder in a polishing machine (Tavom, Wigan, UK).

### 2.3. Sorption and Solubility Test Procedures

For water sorption and solubility experiments, a brass mould was used to prepare specimens with dimensions (50 ± 1.0) mm in diameter and a thickness of (0.5 ± 0.1) mm. Seventy-two acrylic resin specimens were prepared according to EN ISO 20795-1:2008 [12]. Specimens stored in a desiccator containing fresh silica gel were placed in an incubator at (37 ± 1) °C for (23 ± 1) hr and transferred to a second desiccator maintained at room temperature (23 ± 2) °C for 1 h. The specimens were then weighed to an accuracy of 0.0002 g using a calibrated electronic analytical balance (Ohaus Analytical plus, Ohaus Corporation, Parsippany, NJ, USA). The weighing was repeated until the mass loss of each specimen was not more than 0.2 mg in any 24 h period to achieve constant mass in the samples. The final mass in gm was recorded as m1. Each specimen diameter was measured thrice using an electronic digital calliper (Draper, Eastleigh, Hants, UK) to calculate the mean values. The mean of five thicknesses measured at four equally spaced points around the circumference was calculated. The volume (V) of each sample was calculated in cm^3^ by using the mean dimeter and thickness.

For each group, six specimens were immersed in DW and six specimens were immersed in AS at (37 ± 1) °C and stored in an incubator for a period of 180 days. The AS solution was prepared according to a composition shown in Table 1 by dissolving it in 1000 mL of distilled water with a pH value of 5.52 and mixing it using a magnetic stirrer until the components were completely dissolved [17]. After each period (1, 2, 3, 4, 5, 6, 7, 14, 21, 30, 60, 90, 120, 150 and 180 days), the specimens were removed from the storage medium, wiped with a clean dry hand-towel until free from visible moisture, and mass in gm was recorded as m2. After 180 days, specimens were reconditioned in desiccators until a constant mass was reached. This desorption process was carried out over a period of 90 days. The reconditioned mass in gm was recorded as m3.

The values of water sorption (w_sp_) in g/cm^3^ were calculated by Equation (1) [12].
(1)$$w_{\mathrm{sp}}=\frac{m2-m3}{v}$$

The values of water solubility (w_sl_) were obtained in g/cm^3^ and were calculated by Equation (2):(2)$$w_{\mathrm{sl}}=\frac{m1-m3}{v}$$
where m1 is the mass of the sample before immersing in DW or AS, m2 is the mass of the sample after immersing in DW or AS, m3 is the reconditioned mass of the sample after immersing in DW or AS and after a desorption process for 90 days and v is the volume of the specimen in cm^3^ [12]. After completing the calculations, the values for water sorption and solubility were converted to µg/mm^3^.

The percentage mass gain due to sorption and the percentage mass loss due to solubility, which represent the total mass loss of components, were calculated by Equations (3) and (4) [8].
(3)$$\text{Mass gain}\;(\text{sorption}){\;W}_{\mathrm{SP}}\;\left(\%\right)=\;\frac{m2\;\left(t\right)-\;m1}{m1}\;\times 100$$
(4)$$\text{Mass loss}\;(\text{solubility}){\;W}_{\mathrm{SL}}\left(\%\right)=\frac{m1\;\left(t\right)-\;m3}{m1}\times 100$$

### 2.4. Statistical Analysis

The recorded data of sorption and solubility were calculated and statistically analysed using SPSS statistics version 23 (IBM, New York, NY, USA). Non-significant Shapiro–Wilk test showed that the data of sorption and solubility tests were normally distributed and there was homogeneity of variances. The percentage of mass change for sorption and solubility data were analysed with repeated measurements (one-way ANOVA) to evaluate the effect of storage time. The mean values of sorption to solubility among different groups were compared using two-way ANOVA followed by Tukey post-hoc analysis for multiple comparisons at a pre-set alpha value level of 0.05.

## 3. Results

All experimental groups showed that mass changed with time upon storage in DW and AS. For DW, the mass steadily increased in the first 21 days of immersion for all groups. This was then followed by a reduction in the rate of increase until 60 days, and then a slow or even decrease in mass change until equilibrium was reached after 90 days. This was then followed by a slow (except for the control group) increase for the next 60 days, and finally mass was decreased at 180 days (Figure 1). The highest rate of mass increase was with the group containing 3 wt.% of zirconia, but it was not significantly different to other groups.

In AS, all the groups demonstrated a gradual mass increase within the first 7 days except the group containing 10 wt.% of zirconia whose mass was not significantly increased until day 14, as well as showing a significant difference compared to the control group. Mass change for all the groups then decreased until day 21 and then started to increase until day 90. After that, there was no significant mass change (with the course ending on day 180), which was an indication of reaching equilibrium state (Figure 2).

The percentage increase in mass change after 180 days for all the experimental groups in AS was significantly higher than that in DW. However, it should be noted that after 90 days there was no significant change in mass throughout the final 90 days for both solutions. Therefore, sorption values were only presented here for 90 days.

The mean and standard deviation values for sorption and solubility in DW and sorption and solubility in AS are presented in Table 2 and Table 3, respectively. The results showed that at 90 days, sorption by the experimental groups containing zirconia was lower than that by the control group for both solutions, but not significant (*p* > 0.05) (Figure 3).

At 180 days in DW, all groups of nanocomposites showed lower solubility compared to the control group, however, the difference was not significant. The lowest solubility was recorded for the group with 3 wt.% of ZrO_2_ in both solutions. In general, the specimens stored in AS showed significantly lower solubility than specimens stored in DW, as shown in Figure 4. Specimens immersed in DW exhibited a greater colour change compared to the specimens immersed in AS after 180 days and specimens in the dry condition, as shown in Figure 5. However, no visible degradation of the surface after immersing in DW and AS for 180 days was noticed.

## 4. Discussion

In this study, water sorption and solubility characteristics of HI PMMA nanocomposites with different concentrations of ZrO_2_ were measured after being stored for up to 180 days in DW and AS. DW and AS were used as storage media in this study because they simulate oral fluids and react with the denture base inside and outside (stored overnight in water) the mouth. The experimental groups containing zirconia showed reduced sorption in both media compared to the control group, especially with the group containing 3 wt.% zirconia. As the decrease was not statistically significant (*p* > 0.05), the first hypothesis was accepted. The lowest solubility was found for the group containing 3 wt.% of ZrO_2_ in both storage media, after being stored for 180 days but not significantly different from the control group. However, the group containing 10 wt.% of ZrO_2_ was increased significantly different from the control group, and thus the second hypothesis was rejected.

Previous studies reported values of water sorption ranging from 11.64 µg/mm^3^ to 38.31 µg/mm^3^ for different types of denture base acrylic resins after 7 days of immersion in water [5,9,10,14]. The results of this study indicated that HI PMMA nanocomposite experimental groups complied with the British Standard (EN ISO) specification No. 20795 (not exceeding 32 µg/mm^3^ for heat-cured acrylic resin) as mean values ranged from 24.80 µg/mm^3^ to 25.90 µg/mm^3^ compared to the control group (27.05 µg/mm^3^) in DW storage. The mean values of specimen sorption stored in AS ranged from 23.61 µg/mm^3^ to 25.41 µg/mm^3^ after 90 days, compared to the control group (25.17 µg/mm^3^). Therefore, the amount of sorption in both DW and AS is below the standard and lower than the control group. These results indicated that the addition of nanoparticles in HI PMMA did not affect its sorption property negatively, possibly due to a favourable interaction between the particles and the matrix and it is expected that the degradation would be minimal. Furthermore, the British Standard does not specify the kind of degradation and any measurement or estimation of degradation after sorption or solubility.

The findings of this study on water sorption are in agreement with those of Asar et al. who evaluated the effect of reinforcing conventional heat-cured acrylic resin with various metal oxide microparticles (TiO_2_, ZrO_2_ and Al_2_O_3_ using 1% and 2% by volume) on mechanical and physical properties such as, water sorption and solubility [7]. The results showed that incorporation of metal oxides significantly decreased water sorption and solubility, particularly with 2% of ZrO_2_, which showed the lowest solubility compared to the conventional acrylic resins.

In the present study, the lower water sorption could be related to the use of long curing polymerisation (cycling for 6 h in a water bath) to avoid any voids and obtain full polymerisation. Garcia et al, reported that high-temperature curing of acrylic resin increased the degree of conversion that led to an increase in crosslinking, resulting in reduced water sorption values [18]. Dogan et al. studied the effect of the curing time of heat-cured resins on water sorption and concluded that longer curing times led to a reduction in residual monomer content and water sorption [19]. Wong et al. reported that the initial water content in the specimens of denture acrylic resin that were wet heat–cured was slightly lower than that of dry heat–cured (without water bath) [20] due to an increased degree of conversion or a reduced residual monomer content.

When the control group was compared with the groups containing ZrO_2_, it was noted that reinforcement with zirconia nanoparticles led to a decrease in sorption values in DW and AS. The group containing 3 wt.% ZrO_2_ showed the lowest value of sorption after storage in DW compared to the control group, even though the effect was not significantly different. However, the group with 10 wt.% of ZrO_2_ stored in AS demonstrated significantly lower sorption values, followed by the groups with 5 wt.% and 3 wt.% of ZrO_2_ when compared to the control group. Inorganic metal oxide fillers usually demonstrate high surface energy because of their hydrophilic nature. However, due to the difference in surface energies, the hydrophobic polymer does not wet or interact with the filler. It is therefore necessary to alter the surface of the fillers to improve wetting and adhesion at the filler–matrix interface [7], as well as to ensure even dispersion of the fillers throughout the polymer matrix [21]. The decrease in sorption by the nanocomposites in this study may be related to the silane couple agent used for treating the zirconia surface to enhance the interfacial bonding between the zirconia and PMMA matrix. However, the higher surface energy of zirconia could encourage bacterial adhesion, which can be minimised by biofunctionalization [22]. This could be a topic of further research. In the clinical situation, silica coating is also applied to reinforcing particles to increase the hydrophobicity and to enhance bond strength to resin [23]. Furthermore, it was demonstrated that dispersion stability of ZrO_2_ particles can be enhanced by applying a continuous and uniform coating with silica [24]. In the future, the effect of particle coating with silica can be explored. In our previous study [25], it was also observed that the zirconia nanoparticles were fairly distributed within the PMMA matrix without observable particle clustering and the nanoparticles would presumably fill the empty spaces, which indicated a good interaction between the particles and the matrix for improved strength characteristics. Polat et al. mentioned that denture base polymers reinforced with short glass fibres exhibited decreased water sorption and solubility. The reason for this decrease could be attributed to good bonding between glass fibres conditioned by silane coupling agent and the polymer matrix [6].

Other factors that could potentially influence the sorption and solubility of resin composites include storage media, immersion period [8], or filler loading [26]. Water sorption into a polymer is a process when the water is being absorbed and at the same time the polymer material is diffused [27]. Water molecules can spread across polymer matrix due to the small size of the water molecule—the diameter of the water molecule is less than 0.28 nm—which is smaller than the polymer chain distance in the polymer matrix [6].

In this study, according to mass change as shown in Figure 1 and Figure 2, a general trend of gradual increase in mass change over time was found and reached an equilibrium state after 90 days. However, it is interesting to note that for all specimens stored in AS, the mean mass values dropped during 7 days and 14 days, whereas for DW no such trend was observed. The reason for a decrease in mass change values in AS during the first two weeks in our study could be related to AS, a stronger solvent, which contains a mix of various salts and other additives compared to the DW and reacts with the specimen [28]. Therefore, more monomers can come out from the specimens which leads to a decrease in mass and a decrease in sorption in AS. Jagger et al. found a linear relationship between residual monomer and water sorption, which means, as the residual monomer content in the specimens increased, the water sorption also increased [29]. Dogan et al. have hypothesised that the residual monomer near the surface of the resin leaches into water, causing a decrease in polymer mass. If the residual monomer is not close to the surface of the specimen, but rather entrapped in the inner layers of resin specimen, it cannot diffuse into water after a short time [19].

The highest mass change was observed for the control group and the lowest for the group containing 10 wt.% zirconia for all of the test durations. This could be related to the fact that the addition of zirconia fills the empty spaces in the PMMA resin and reduces sorption of saliva or mass change compared to the control group. Alsharif et al., who studied the effect of different ratios of filler loading on saliva sorption and solubility in resin composite, found that samples with higher filler loadings showed the lowest change in mass when compared to the unfilled resin. They explained this as a result of filler loading [30]. In the current study, the changing of mass during long-term storage may be related to the release of residual monomer which is, at the same time, replaced with storage solution, leading to an increased sorption, particularly with the control group. However, the inclusion of ZrO_2_ might reduce the amount of residual monomer in the nanocomposite composition, evidenced by less sorption in comparison to the control group.

The current study found that sorption in DW was higher than that in AS. When the water diffuses into the material it is pulled to the hydrophilic/water dissolvable sites and solution droplets are formed. The contrast in osmotic pressure between the inside solution droplet and the outside solution will result in an osmotic force to increase the sorption [28].

Ergun et al. investigated the effect of reinforcing conventional heat-cured PMMA with concentration of (5 wt.%, 10 wt.% and 20 wt.%) zirconia nanoparticles on water sorption and solubility after 28 days [31]. The findings showed that an increase in ZrO_2_ concentration increased water sorption. This was explained by the weak polymer chains allowing water to permeate into the matrix and the increase in the nanoparticle filler concentration resulting in more nanoparticles at the filler–matrix interface [31]. Kundie et al. evaluated the effect of different ratios of nano- and micro-alumina fillers on water sorption and solubility in denture base acrylic resin after 7 days. The reinforced samples showed slightly higher water sorption than the control group. The reason was attributed to the filler particle size and distribution and interfacial bond between the filler and the resin matrix [32]. Al-mulla et al. investigated the effect of HI acrylic resin, conventional heat-cured acrylic resin and other types of resin on sorption over 28 days storage in two types of artificial saliva and distilled water. They found that the HI resin had slightly less sorption in distilled water than that in artificial saliva, but not significantly. Whereas in conventional resin, sorption in distilled water was higher than that in the artificial saliva [27]. The findings of these previous studies contradict with the results obtained in the present study, in which all groups containing (1.5, 3, 5, 7, 10 wt.%) zirconia exhibited lower sorption values compared to the control group in both storage media.

Solubility represents the mass of the materials dissolved from the polymer. The only soluble materials present in denture base resins are residual or unreacted monomers, plasticisers and initiators that are most likely leached over the immersion time [13]. This study showed that the lowest mean value of solubility was found in the group containing 3 wt.% of ZrO_2_ stored in AS (−3.03 ± 0.95 µg/mm^3^), whereas the mean value of the control group was −2.32 ± 0.89 µg/mm^3^, both of which were lower than the British Standard (EN ISO) specification No. 20795 (solubility value should not exceed 1.6 µg/mm^3^ for heat-cured acrylic resin). However, the groups stored in DW had higher solubility compared to the groups stored in AS and the recommended value by the British Standard. The highest mean value of solubility was 8.01 ± 5.48 µg/mm^3^ in the control group and lowest mean value of 3.78 ± 2.64 µg/mm^3^ was for the group containing 3 wt.% ZrO_2_.

The solubility values obtained in this study for specimens stored in DW were higher than those reported in previous studies (ranging from 0.03 to 2.68 µg/mm^3^) [5,7,10,13,14]. However, specimens stored in AS showed lower solubility values compared to the earlier mentioned studies that evaluated the solubility of different types of acrylic resin. This difference in values could be attributed to the amount of water sorption over a long time being high enough to offset the mass lost by the diffusion of residual monomer [19]. In addition, Silva et al. highlighted that an increase in the amount of water absorbed could lead to an increase in the capacity for dissolution [33]. Pfeiffer et al. reported that water sorption and solubility of polymers depended on the homogeneity of the material. The more homogeneous a material is, the less water it absorbs and the less soluble it is [14]. In the present study, this might be the reason for higher solubility in DW compared to that in the artificial saliva.

To overcome the limitations of the current in vitro study, further investigation should be designed to evaluate residual monomer and thermal expansion for HI PMMA nanocomposite denture base using artificial saliva to provide additional simulation of the oral environment more accurately. The specimen size should be increased to 10 specimens per group in order to obtain a better statistical distribution.

## 5. Clinical Implications

The HI PMMA nanocomposite denture base impregnated with less than 10 wt.% of zirconia did not make any significant difference in sorption and solubility for a long period of time (180 days) compared to the control group.

## 6. Conclusions

Within the limitations of this in vitro study, the following conclusions can be drawn. HI PMMA resin reinforced with zirconia nanoparticles with a range of concentrations (1.5, 3, 5, 7 and 10 wt.%) showed lower sorption compared to the control group (0 wt.% zirconia) in both distilled water (DW) and artificial saliva (AS) for up to 90 days but was not statistically significant. All nanocomposite groups in DW showed lower solubility than the control group after 180 days of storage but was not statistically significant. However, the nanocomposite with 10 wt.% of ZrO_2_ showed significantly higher solubility in AS compared to the control group. Therefore, a low concentration of zirconia (1.5 to 5 wt.%) in PMMA will not significantly affect the sorption and solubility properties of the nanocomposite.

## Figures and Tables

**Figure 1 materials-13-03732-f001:**
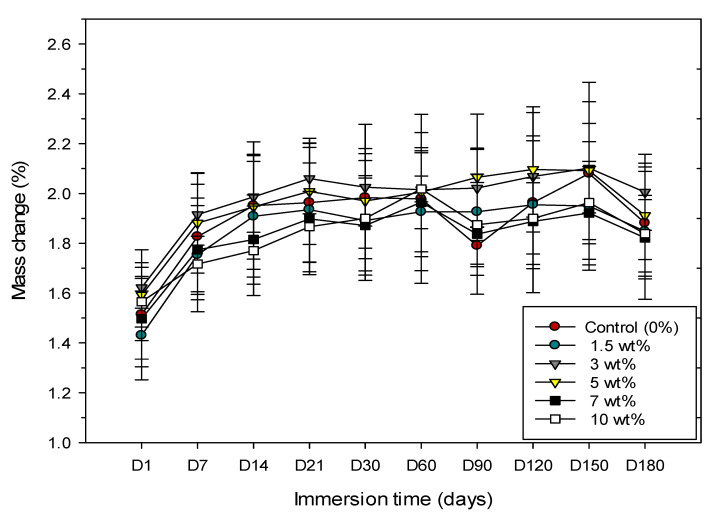
Line graph illustrating the mass change of specimens from six groups immersed in distilled water over 180 days.

**Figure 2 materials-13-03732-f002:**
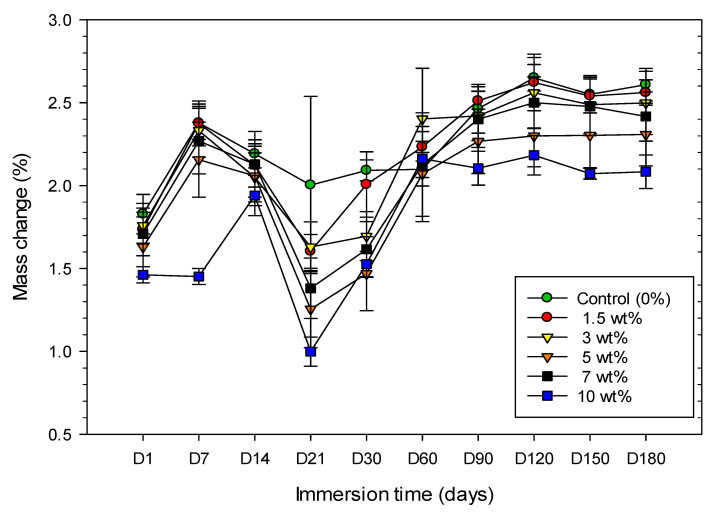
Line graph illustrating the mass change of specimens from six groups immersed in artificial saliva over 180 days.

**Figure 3 materials-13-03732-f003:**
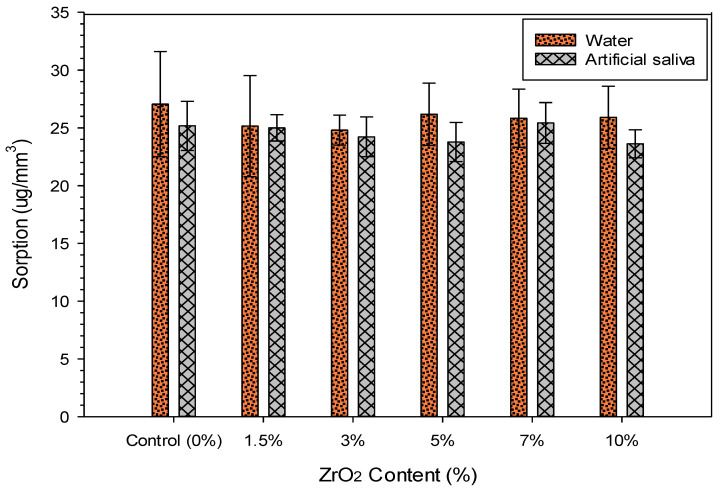
Bar chart demonstrating the mean solubility values of specimens after immersing for 180 days in distilled water and artificial saliva.

**Figure 4 materials-13-03732-f004:**
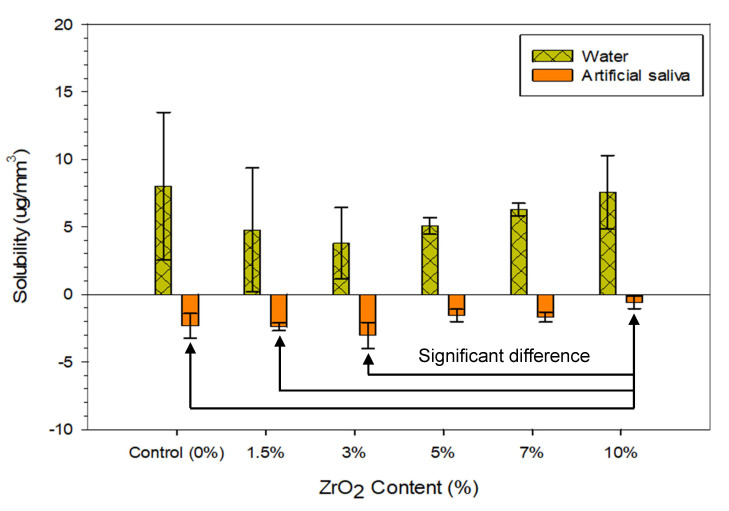
Bar chart demonstrating the mean solubility values of specimens after immersing for 180 days in distilled water and artificial saliva.

**Figure 5 materials-13-03732-f005:**
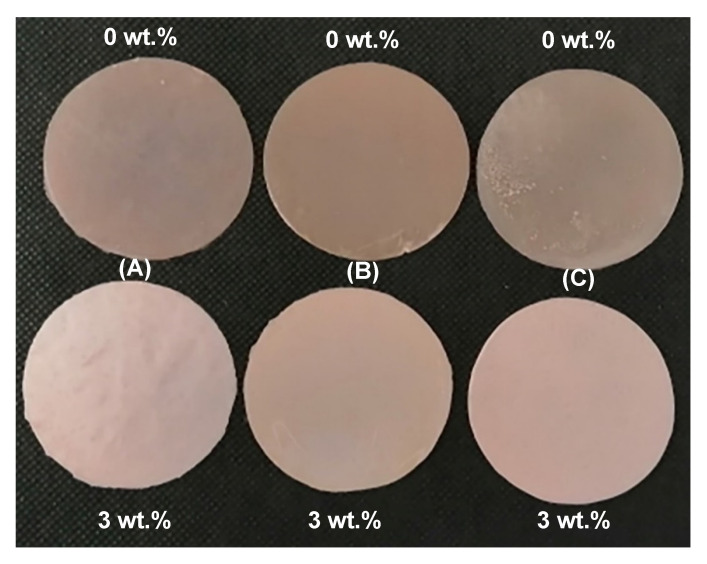
Photograph showing the colour changes in the control group (0 wt.%) and group containing 3 wt.% of ZrO_2_ (**A**) before immersion in distilled water (DW) or artificial saliva (AS), (**B**) after immersing for 180 days in DW and (**C**) after immersing for 180 days in AS.

**Table 1 materials-13-03732-t001:** Composition of artificial saliva [17].

Compound	Amount (g/L)	Manufacturer	CAS
Sodium chloride (NaCl)	0.400	Acros Organics	7647-14-5
Potassium chloride (KCl)	0.400	Fisher Chemical	7447-40-7
Calcium chloride (CaCl_2_)	0.795	Acros Organics	10043-52-4
Sodium dihydrogen phosphate (H_2_NaO_4_P)	0.690	Alfa Aesa	7558-80-7
Sodium sulphide hydrate (Na_2_SxH_2_O)	0.005	Acros Organics	27610-45-3

**Table 2 materials-13-03732-t002:** Mean and standard deviation values of sorption (µg/mm^3^) and percentage of sorption (%) after 90 days of storage in distilled water; solubility (µg/mm^3^) and its percentage (%) values after 180 days.

Experimental Group	Water Sorption (%)	Water Sorption (µg/mm^3^)	Water Solubility (%)	Water Solubility (µg/mm^3^)
Control 0%	1.79 (0.07) ^a^	27.05 (4.55) ^a^	0.75 (0.52) ^a^	8.01 (5.48) ^a^
1.5%	1.92 (0.25) ^a^	25.16 (2.00) ^a^	0.45 (0.43) ^a^	4.78 (4.57) ^a^
3%	2.02 (0.15) ^a^	24.80 (1.29) ^a^	0.36 (0.25) ^a^	3.78 (2.64) ^a^
5%	2.06 (0.25) ^a^	26.18 (2.68) ^a^	0.42 (0.18) ^a^	5.06 (0.60) ^a^
7%	1.83 (0.24)^a^	25.83 (2.51) ^a^	0.59 (0.04) ^a^	6.28 (0.46) ^a^
10%	1.87 (0.16) ^a^	25.90 (2.69) ^a^	0.71 (0.25) ^a^	7.55 (2.71) ^a^

Note: Same lower-case letters within column indicate no significant difference (*p* > 0.05) from control.

**Table 3 materials-13-03732-t003:** Mean and standard deviation values of sorption (µg/mm^3^) and percentage of sorption (%) after 90 days of storage in artificial saliva; solubility (µg/mm^3^) and its percentage (%) values after 180 days.

Experimental Group.	Saliva Sorption (%)	Saliva Sorption (µg/mm^3^)	Saliva Solubility (%)	Saliva Solubility (µg/mm^3^)
Control 0%	2.46 (0.14) ^a^	25.17 (2.12) ^a^	−0.20 (0.07) ^a^	−2.32 (0.89) ^a^
1.5%	2.51 (0.08) ^a^	24.99 (1.13) ^a^	−0.22 (0.02) ^a^	−2.40 (0.28) ^a^
3%	2.42 (0.14) ^a^	24.22 (1.72) ^a^	−0.26 (0.07) ^a^	−3.03 (0.95) ^a^
5%	2.26 (0.19) ^a^	23.77 (1.68) ^a^	−0.14 (0.04) ^a,b^	−1.56 (0.46) ^a,b^
7%	2.40 (0.16) ^a^	25.16 (1.76) ^a^	−0.14 (0.03) ^a,b^	−1.68 (0.36) ^a,b^
10%	2.10 (0.10) ^a^	23.61 (1.20) ^a^	−0.05 (0.03) ^b^	−0.58 (0.46) ^b^

Note: Same lower letters within column indicated no significant difference (*p* > 0.05) from control.

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
