# Peer review of "Long-Term Sorption and Solubility of Zirconia-Impregnated PMMA Nanocomposite in Water and Artificial Saliva"

_materials, 2020, doi:10.3390/ma13173732_

Round 1

Reviewer 1 Report

Authors present a very interestiing paper on the in vitro study  to assess the sorption and solubility of high-impact heat-polymerized denture base PMMA impregnated with zirconia nanoparticles after storing for 180 days in distilled water and artificial saliva.

The aim is to reduce  the loos of physical properties and  clinical service life.

The paper is well presented and developed, with a correct introduction and state of art. The experimental procedure is well explained and performed, according to standards for clinical use. I believe that obtained conclusions are in agreement with obtained experimental data, validating the scientific procedure and obtaining interesting results.

However, I find some corrections that have to be made. Firstly, the abstract has to underline the important aspects of the paper and the relevant results and conclusions, but not to show how the experimental procedure was developed. It must be rewritten for a better understanding of the main highlights of the paper.

Moreover, Figures 1 and 2 are quite confussing, as too many symbols and error bars are present, difficulting the understanding of the figures. I suggest to rearrange both figures.

Finally, I suggest to make conclusions shorter and not only describe the results obtained for composites....it would strenngth the paper value, in my opinión.

Author Response

Hi Dear Reviewer 1

Reviewer 2 Report

This is a well-written paper. The results were presented; however, I have some questions as below:

Material and method section:

1) Exact values (mean and SD) of the base mass (m1) for specimens in control and the experimental group should be reported. Mass of the denture is also one of the important factors since it affects the retention of the denture, particularly in a complete denture. One of my concerns is that the incorporation of the zirconia may increase the mass of the denture. The issue should be discussed.

2) Line 141: “The values of water solubility (wsp)” should be changed to “The values of water solubility (wsl)” as the annotation in the bellowed equation. The equation (3): change in mass SP was ambitious, do you mean the gain in mass due to the water sorption? For the reader’s convenience, annotation of equations should be written following the respective equation.

Result section:

3) Table 2 shows higher water sorption in the percentage of the experiment group than the control group, while the water sorption in µg/mm3 shows the inversed trend. Please explain the issue.

4) Table 3 shows a significant difference between some of the experimental groups and the control group in water solubility. Please make annotations to the bar chart in Figure 4 to show the significant difference.

Discussion section:

5) The incorporation of ZrO2 may affect the surface energy of the restoration, thus, it also affects the bacterial adhesion. The fact should be further discussed.

6) In the clinical situation, silica coating and silane surface treatment are often used to increase the hydrophobicity of the acrylic denture. As the coating procedure was not conducted in this study, the fact should be explained and discussed further.

7) Line 208, the authors stated that the lowest mass change was observed for the group containing 10 wt.% zirconia for all of the test durations; however, in clinical implications, the authors have suggested that the HI PMMA nanocomposite denture base impregnated with 3 wt.% of zirconia will have the potential to replace traditional PMMA with increased clinical life of artificial denture. What is the advantage of using 3 wt.% of zirconia instate of 10 wt.% zirconia? Please explain and discuss further the fact.

Conclusion section:

8) Line 208 the authors stated that: The highest mass change was observed for the control group and lowest for the group containing 10 wt.% zirconia for all of the test durations. However, in conclusion, the authors concluded that the nanocomposite with 10 wt.% of ZrO2 showed significantly higher solubility in AS compared to the control group. It may confuse the readers, please revise the sentences.

Author Response

Hi Dear Reviewer 2

Reviewer 3 Report

I have read the article “Long-Term Sorption and Solubility of Zirconia-2 Impregnated PMMA Nanocomposite in Water and 3 Artificial Saliva” with interest. Authors touch on the interesting problem of long-term sorption and solubility, for PMMA polymeric matrix. This remains an interesting area to pursue to study. However, their overall motivation is not justified sufficiently.

At the moment – although interesting, the study lacks very basic information about the surface modification after immersion in both medium

The interactions between the Zirconia-2 and matrix are not clear. Furthermore, these are not all linked to the actual needs of the field (i.e. what kind of degradation/ what are current standards?).  I kindly suggest Authors to consider these points and undertake minor revisions of the manuscript before it can be deemed publishable.

In the Materials chapter, the company, city, country must be indicated

Instead of figure 5 I would indicate, electronic scanning microscopy, SEM, with comparison between the initial samples, before immersion and after immersing for 180 days in both medium (water and artificial saliva).

Figure 5 was indicated, if the color tests were performed, before and after immersion in the two environments

Author Response

Hi Dear Reviewer 3
